# ON THE LIMITS OF CURRICULUM LEARNING FOR POST-TRAINING LARGE LANGUAGE MODELS

## ABSTRACT

Large language models (LLMs) excel at many common-sense tasks, yet they remain brittle when required to perform consistent multi-step reasoning. Evaluations on benchmarks such as AMC or AIME25 are often affected by data contamination, motivating our focus on synthetic reasoning tasks with controllable difficulty. Synthetic datasets allow us to generate problems whose difficulty directly corresponds to the number of (verbalized) reasoning steps required. By focusing on synthetic tasks with minimal natural language complexity, we ensure that our conclusions are driven by reasoning ability rather than sophisticated linguistic understanding. We investigate generalization to higher difficulty levels at the granularity of individual difficulties, a setting that differs from the standard out-of-distribution evaluation, which typically tests on entirely different tasks. To improve generalization to harder problems, we study curriculum learning (CL) as a mechanism to exploit difficulty during post-training. Across multiple synthetic reasoning tasks and a family of medium-sized models, we find that CL has no significant impact under either supervised fine-tuning (SFT) or reinforcement learning (RL). Moreover, the optimal CL schedule varies across datasets and models, while standard random sampling performs competitively. We identify response length as a key factor driving model performance, and observe that CL schedules do not significantly impact response length, explaining why SFT performance does not improve with CL. While pre-training commonly adopts data mixing strategies akin to curriculum learning, these findings call into question the usefulness of curriculum learning for post-training in mathematical reasoning tasks, and suggest that future work should explore alternative mechanisms for strengthening pure reasoning robustness in LLMs.

## 1 INTRODUCTION

Large language models (LLMs) have achieved remarkable success on a wide range of natural language understanding and generation tasks (Brown et al., 2020; Raffel et al., 2020; Touvron et al., 2023). However, while they often excel on common-sense and factual questions, their performance on tasks requiring consistent multi-step reasoning remains far from robust (Kojima et al., 2022; Wei et al., 2022). Improving reasoning ability has therefore become a major research direction, with many approaches relying on *post-training* methods such as supervised fine-tuning (SFT) with chain-of-thought (CoT) traces (Wei et al., 2022; Ho et al., 2023), or reinforcement learning (RL) methods like PPO and GRPO that optimize responses via hand-crafted reward functions or LLM-based judges (Ouyang et al., 2022; Rafailov et al., 2023). Both families of methods encourage models to reason explicitly before producing an answer: in SFT by training on CoT traces, and in RL by asking the model to think first and then produce a final answer in its response.

Mathematical and logical reasoning benchmarks such as GSM-8K (Cobbe et al., 2021a), MATH (Hendrycks et al., 2020), and AIME (Balunović et al., 2025) have become the standard evaluation suite. Yet, these datasets suffer from *data contamination*: many problems appear verbatim or in paraphrased form in pre-training corpora of modern LLMs (Sainz et al., 2024; Wu et al., 2025). Even AIME25 includes problems available online prior to 2025, making it unreliable as a measure of true generalization of LLMs with cutoff dates prior to 2025. This has led to surprising results, such as reinforcement learning seemingly succeeding from only a handful of examples or even from random rewards (Wang et al., 2025; Shao et al., 2025). Closer inspection often reveals that such

effects stem from contaminated benchmarks (Wu et al., 2025), misreported baselines, or overly permissive evaluation protocols (Chandak et al., 2025). To avoid these confounding effects, we focus on *synthetic mathematical reasoning* datasets without the risk for data contamination, and where the difficulty can be systematically controlled and is indicative of the required reasoning depth. Examples include MATHGAP (Opedal et al., 2025), KNIGHTS-AND-KNAVES puzzles (Xie et al., 2025; 2024), ZEBRALOGIC (Lin et al., 2025), RANDOMCALCULATION (Wu et al., 2025), and we focus on three such datasets.

In this work, we investigate whether *curriculum learning* (CL)—the strategy of ordering training data by difficulty (Bengio et al., 2009; Hacohen & Weinshall, 2019)—improves post-training performance on synthetic mathematical reasoning tasks. *Reasoning* requires applying simple rules repeatedly to form longer reasoning chains. It is known that performance typically degrades with reasoning length (Anil et al., 2022), but not specifically for reasoning and whether this can be improved by CL. Intuitively, one might expect that training on shorter problems first would bootstrap the ability to solve longer problems, making better use of compute and paralleling educational curricula. We start by studying zero-shot performance on synthetic mathematical reasoning tasks. We then use curriculum learning (CL) as an instrument to assess whether CL improves in-distribution and out-of-distribution performance. Since the generated problems follow the same structure, we define out-of-distribution generalization as the ability to transfer the performance on easy examples to more difficult examples, thereby focusing on consistency of learned rules rather than the typical out-of-distribution generalization on entirely different tasks. Since CL acts on difficulty, there is reasonable expectation that it can improve performance on more difficult examples. We test this hypothesis by systematically evaluating curriculum schedules across both SFT and RL fine-tuning. Our results suggest that curriculum learning has little to no benefit in this setting, raising questions about its utility for post-training on synthetic mathematical reasoning tasks. We make the following **contributions**:

- We evaluate zero-shot performance of LLMs, showing that our notion of difficulty correlates with LLM performance.

- We compare the effect of various curriculum schedules during post-training on synthetic reasoning tasks from the angles of accuracy, response length and format-following, distinguishing between in-distribution and out-of-distribution. We find that none of the schedules we consider consistently improves performance, despite the common use of CL in pre- and post-training.

- We compare RL and SFT, pinpoint reasoning length as a possible cause for SFT breakdown, and discuss the differences in the light of the paradigm "SFT memorizes, RL generalizes" (Chu et al., 2025). We identify scenarios where SFT loses its mathematical abilities, whereas RL improves them.

## 2 RELATED WORK

**Post-training methods.** Adapting pre-trained large language models (LLMs) to downstream tasks is a central step in modern LLM development. Two dominant paradigms are supervised fine-tuning (SFT) and reinforcement learning (RL). SFT, both with and without chain-of-thought (CoT) traces, has been successfully applied to align LLMs with task-specific behaviors and reasoning styles (Wei et al., 2022; Ouyang et al., 2022). By contrast, RL-based approaches, such as reinforcement learning from human feedback (RLHF), are significantly more expensive, as they require on-policy rollouts and reward modeling (Christiano et al., 2017; Ouyang et al., 2022). Recent work has proposed more efficient RL variants; for example, RL with Verifiable Rewards (RLVR) focuses on outcome-based reward signals, and has been successfully applied to reasoning and code generation tasks (Yu et al., 2025; Shao et al., 2024). Post-training thus remains an active area of research, with trade-offs between performance, efficiency, alignment quality, and scalability (Tie et al., 2025).

**Reasoning in LLMs.** Recent work has highlighted the limitations of LLMs in multi-step reasoning and proposed various methods to address them (Wei et al., 2022; Kojima et al., 2022; Sprague et al., 2023). Chain-of-thought (CoT) prompting (Wei et al., 2022; Kojima et al., 2022) and fine-tuning with reasoning traces (Wei et al., 2022; Ho et al., 2023) improve performance by encouraging explicit intermediate steps. RL-based methods such as PPO and GRPO (Ouyang et al., 2022; Rafailov et al.,

2023) instead rely on reward shaping, often incorporating response format scores, or LLM-based judge models (Ouyang et al., 2022; Cobbe et al., 2021b). While SFT is often effective at incorporating new knowledge, RL is argued to better promote generalization within existing knowledge (Ma et al., 2025; Chu et al., 2025). Chu et al. (2025) showed that RL generalizes better than SFT on an arithmetic and spatial reasoning task, but did not study the effect of curriculum learning and did not study generalization with respect to the number of reasoning steps. We consider a large range of difficulties/steps. Compositional generalization — the ability to solve more complex problems by composing the solutions to simpler subproblems — is closely related to our notion of generalization, and is an active research area (Wiedemer et al., 2023; Li et al., 2024; Zhao et al., 2025).

**Data contamination and spurious results.** A growing body of work has shown that standard math datasets are heavily contaminated in modern LLM pre-training corpora, leading to unreliable evaluations (Xu et al., 2024; Frieder et al., 2024). For instance, AIME-2024, MATH-500, AMC are all confirmed to appear in the training data of models such as the Qwen2.5 (Team, 2025) and Qwen3 (Yang et al., 2025) series (distilled from Qwen2.5) (Wu et al., 2025). Even AIME25 contains problems accessible online before the official release (Papailiopoulos, 2025). This can happen even when not directly fine-tuning on the dataset, but distilling on text generated by another contaminated LLM, as text can contain hidden messages (Cloud et al., 2025). This undermines claims about reasoning ability and makes it difficult to separate memorization from genuine generalization. Mirzadeh et al. (2024) and Xie et al. (2024) both showed that performance degrades significantly when substituting numbers in the question, hinting at memorization. Wu et al. (2025) gave strong arguments that the claims made in Shao et al. (2025); Wang et al. (2025) were caused by data contamination. This was further hinted by the fact that these results only seemed to hold on the Qwen model series, but not on Llama models.

**Real-world and synthetic reasoning benchmarks.** While the mentioned real-world datasets provide challenging problems of varying difficulty, they suffer from data contamination (as explained above) and lack systematic control over reasoning complexity. Therefore, synthetic benchmarks have been proposed. Chen et al. (2025); Liu et al. (2023); Han et al. (2022); Tafjord et al. (2020) propose benchmarks focusing on logical tasks with complex natural language to imitate real-world data. We instead target synthetic reasoning benchmarks without complex natural language to test the compositional reasoning ability of LLMs, minimizing confounding due to natural language complexity. We rely on existing synthetic datasets that use simple templating. While this may seem like a limitation for real-world use cases, it helps us focus on reasoning ability and study the effect of curriculum learning, while being sufficiently challenging to justify post-training. Such synthetic benchmarks include MATHGAP (Opedal et al., 2025), KNIGHTS-AND-KNAVES puzzles (Xie et al., 2025; 2024), ZEBRALOGIC (Lin et al., 2025), or RANDOMCALCULATION (Wu et al., 2025). They provide an attractive alternative allowing for controlled generation of reasoning traces at chosen difficulty levels, while avoiding contamination. Many of the synthetic benchmarks only consider the zero- or few-shot performance, whereas we focus on improving it via post-training, and the effect of curriculum learning on this. While LLMs are also trained on synthetic datasets including reasoning tasks, exact memorization is not possible due to the size of the problem space, so re-generating the synthetic dataset avoids contamination.

**Curriculum learning.** Erhan et al. (2009) empirically found that pre-training deep neural networks for a downstream task does not reduce the final training error, but improves the test error, suggesting that pre-training both helps optimization and acts as a regularizer. Exploiting this observation, Bengio et al. (2009) introduced curriculum learning (CL) as a training paradigm where models are exposed to increasingly difficult examples, adding more difficult examples whilst keeping the simple examples. Follow-up explored pacing functions, difficulty annotations, and conditions under which CL improves generalization (Hacohen & Weinshall, 2019; Soviany et al., 2022; Wu et al., 2020). Soviany et al. (2022) point out that CL may help unsupervised learning by implicitly providing an additional signal — the difficulty of the examples — to the model. CL is not limited to increasing difficulty. Sometimes, ordering the examples by decreasing difficulty gives better performance (Zhang et al., 2018; Cirik et al., 2016; Soviany et al., 2022). CL has been applied to a wide range of tasks (Soviany et al., 2022), including computer vision (CV), speech, robotics, natural language processing (NLP), with sometimes mixed findings about which difficulty metric, curriculum schedule, or hyperparameter setting (such as the learning rate) is best (Zhang et al., 2018; Wu et al., 2020; Cirik et al., 2016; Soviany et al., 2022;

Xie et al., 2025). Difficulties can be derived from human annotation, model-derived difficulty prior to training (e.g. zero-shot performance), or constantly updated during the training (Shi et al., 2025; Soviany et al., 2022). Here, we are interested in measuring difficulty based on reasoning complexity, as defined by the underlying reasoning tree structure, see Section 3.1. CL has been widely used in LLM pre-training, training on easier examples first and gradually adding more difficult ones (Brown et al., 2020; Raffel et al., 2020; Mi, 2023; Zhang et al., 2025), be it by adjusting the dataset mix during training or some form of curriculum schedule. CL has been used in post-training as well (Team et al., 2025b), but without comparing different schedules, assuming that the best schedule starts with easy examples. Similarly to CL, Guo et al. (2025) interleave stages of SFT and RL with increasing difficulty. Setlur et al. (2025) increase problem difficulty over training and adapt the response token budget to difficulty. Shi et al. (2025) study an adaptive curriculum schedule based on the model's recent reward signals and show its benefits on common real-world datasets using the contaminated Qwen2.5 model. We focus on comparing different curriculum schedules, using synthetic reasoning tasks to avoid data contamination and uncontrolled difficulty.

## 3 METHODOLOGY

### 3.1 DATASET DESCRIPTION

We use synthetic datasets to systematically assess the effect of CL on arithmetic and logical reasoning in LLMs, while minimizing the risk of data contamination. Let $d$ denote the difficulty level of a problem instance. The datasets all come with chain-of-thought (CoT)/reasoning traces. Tables 1 and 2 present example problems.

| Problem Type | Difficulty | Problem | Reasoning Trace | Answer |
|---|---|---|---|---|
| lindepth | 2 | Joshua owns 15 meters of rope. Joshua then receives 14 more meters of rope from Christopher. Joshua then gives Christian 9 meters of rope. How many meters of rope does Joshua have? | Joshua owns 15 meters of rope. Joshua then receives 14 more meters of rope from Christopher. So Joshua has 15 + 14 = 29 meters of rope. Joshua then gives Christian 9 meters of rope. So Joshua has 29 - 9 = 20 meters of rope. | 20 |
| partwhole | 2 | Joshua owns 13 phones. Christian has 12 phones. If everyone sums up the phones that they have, how many phones does everybody have combined? | Joshua owns 13 phones. Christian has 12 phones. If everyone sums up the phones that they have, there are 13 + 12 = 25 in total. | 25 |

Table 1: Dataset examples. More examples are in Appendix Table 2.

**Arithmetic reasoning.** The MATHGAP dataset (Opedal et al., 2025) generates arithmetic word problems with CoT traces. We consider two subtypes of problems derived from MATHGAP:

- **LINEARDEPTH**: $d$ participants pass integer quantities of objects among each other. Each step introduces a new entity and establishes its relationship to a previously introduced one. The reasoning process corresponds to a proof tree of linear depth $d$ width $d$ (Opedal et al., 2025).

- **PARTWHOLE**: Each of $d$ participants initially possesses a certain quantity, and the task is to compute the total amount across all individuals. The CoT trace is constructed from a proof tree of depth 1 and width $d$, but could equivalently be constructed from a proof tree of depth $d$ and width $d$, adding one number at a time. Here, we pick the former to assess the impact of reasoning style on SFT.

The success rate of a random predictor is vanishingly small, as it has to guess the correct number. Each problem has an underlying proof tree structure that naturally defines difficulty, yielding a total order over problem difficulty. Problems of different difficulty levels are structurally related: mastering easier instances can help with harder ones, and conversely.

**Logical reasoning.** The KNIGHTS AND KNAVES (KK) dataset (Xie et al., 2024) consists of logical puzzles with $d$ characters. Each character is either a knight (which always tells the truth) or a knave (which always lies). The goal is to infer the truthfulness of all characters by analyzing the logical consistency of their statements about one another. This can be formulated as a boolean satisfiability problem (SAT) with possibly multiple solutions. We restrict attention to instances that admit a unique

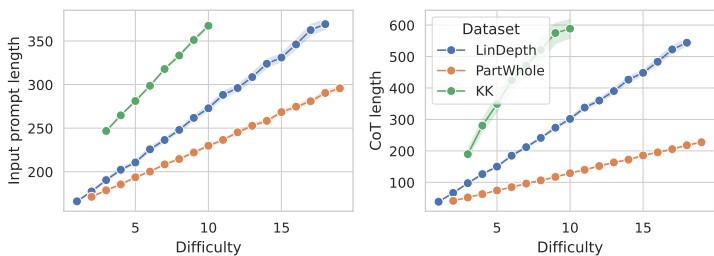

Figure 1: Number of tokens of the input prompt and (ground-truth) CoT trace (reasoning trace and answer). Tokenization is performed with QWEN3-0.6B.

solution. The CoT trace iteratively constructs a role assignment, ruling out impossible assignments by following the logical consistency of the statements, and backtracking when necessary. This task is more challenging than MATHGAP, and requires counterfactual reasoning, carefully ruling out impossible assignments through a series of backtracking steps. It is possible to arrive at the final solution by chance without having to backtrack. The success rate of a random predictor is $2^{-d}$. Brute-force search take $2^d/2$ rounds on average.

**Difficulty and model behavior.** Figure 1 shows the number of tokens of the prompt and the CoT trace. We see that the input prompt and CoT length increase linearly with difficulty. For KK, the CoT length increases more steeply but linearly, pointing to the fact that KK is more challenging than LINEARDEPTH and PARTWHOLE, as it involves significant backtracking. We do not attempt to factor out the correlation between difficulty and input length, but instead analyze the overall effect of difficulty on performance. Although these tasks may appear simple, LLMs often struggle with them in practice: they may enter reasoning loops or reference incorrect quantities. A common error we observe occurs when the result of a prior step coincides with a number introduced in the current step, leading to "confusion" in the reasoning trace.

### 3.2 CURRICULUM LEARNING SETUP

We now formalize the curriculum learning (CL) setting considered in this work. We use the term *curriculum learning* to denote any structured ordering of training examples, not only the canonical "easy-to-hard" progression. Assume a training dataset with difficulty levels $1, \ldots, d$, each containing $n$ samples, so that the total dataset size is $N = nd$. This requires a total order over the dataset, which we have defined systematically in Section 3.1. We train each model for a fixed budget of $(d + r)m$ epochs, where $r \geq 0$ is a constant offset and each epoch consists of $n$ samples. Each curriculum phase lasts $m$ epochs, the last one phase is repeated $r$ times. Fixing the number of samples per epoch is advantageous, since hyperparameters such as the learning rate would otherwise need to be re-tuned (Hacohen & Weinshall, 2019). We evaluate five curriculum schedules, where $//$ denotes integer division:

- *Standard* (*baseline*): Standard random sampling. Each epoch samples $n$ datapoints uniformly across all difficulty levels.
- *SingleDiffInc*: At epoch $i$, sample $n$ datapoints from difficulty $\min(i//m, d)$.
- *SingleDiffDec*: At epoch $i$, sample $n$ datapoints from difficulty $\max(d + 1 - i//m, 1)$.
- *UpToDiff*: At epoch $i$, sample $n$ datapoints from difficulties $1, \ldots, \min(i//m, d)$.
- *DownToDiff*: At epoch $i$, sample $n$ datapoints from difficulties $\max(d+1-i//m, 1), \ldots, d$.

Up to random sampling, each model is exposed to all $N$ datapoints after $n = dm$ epochs, and all curricula involve the same total number of training steps. The *Standard*, *UpToDiff*, and *DownToDiff* curricula all train on the complete dataset after $(d - 1)m$ epochs. In practice, both pre-training and post-training of LLMs often employ variants of *UpToDiff* or *SingleDiffInc*.

To avoid introducing additional hyperparameters, we fix the number of epochs per phase. More complex pacing strategies are possible, such as adapting phase length as a function of difficulty

or training until convergence on a validation metric (which may differ from the downstream task metric) (Wu et al., 2020; Soviany et al., 2022). However, these approaches introduce substantial additional design choices and are therefore not pursued here. Moreover, the problem difficulties can be derived from human annotations, model performance before or during training, or other proxies. Here we focus on reasoning complexity and employ the difficulty levels used during synthetic data generation, thereby eliminating this additional source of variation.

### 3.3 PROMPT FORMAT AND OUTPUT PARSING

We adopt a structured prompt format for both supervised fine-tuning (SFT) and reinforcement learning (RL). The prompts are designed to elicit explicit reasoning traces and a final answer, in line with formats widely used for reasoning tasks (Guo et al., 2025; Shao et al., 2024; Yu et al., 2025; Xie et al., 2025):

```
# SFT
{TASK_DESCRIPTION} {problem} <assistant_start> <think> {reasoning_trace}
    </think> <answer> {answer} </answer>

# RL-finetuning (RFT), zero-shot, and evaluation prompt
{TASK_DESCRIPTION} {problem} <assistant_start> <think>
```

The `TASK_DESCRIPTION` briefly describes the task and the expected output format. For both RL and evaluation, this description explicitly states the expected output format `<think>.*</think><answer>.*</answer>` to parse answers reliably. The SFT prompt additionally provides the reasoning trace, which gives a stronger training signal than RL.

The RL prompt terminates with the `<think>` tag to encourage the model to reflect on the problem before producing its answer. We found this particularly beneficial for adherence to the desired format in preliminary experiments. For the KK dataset, the RL prompt also includes a one-shot example, which we found helpful for stabilizing training.

**RLVR setup.** We adopt the RL with Verifiable Rewards (RLVR) framework (Yu et al., 2025; Shao et al., 2024), which combines multiple reward components:

- **Formatting**: reward for adhering to the required output format (approximately or exactly).
- **Correctness**: reward for producing the correct final answer. For MATHGAP, an additional bonus is given if the answer is an integer.
- **Length**: reward for staying within the response length limit, penalizing truncation.

Following the correct format influences the correctness reward. Appendix C.2 details the weighting of these components for each dataset. Although results can in principle be sensitive to reward shaping, we did not observe substantial effects in practice.

## 4 RESULTS

We first assess the zero-shot performance of various models, for different model families and model sizes. We then study the effect of curriculum learning on RL (GRPO, PPO) and SFT. All metrics are reported on a validation dataset of 128 examples per difficulty. We abbreviate the models as follows: L1B (LLAMA3.2-1B), L3B (LLAMA3.2-3B), Q0.6B (QWEN3-0.6B), Q1.7B (QWEN3-1.7B), Q4B (QWEN3-4B), G9B (GEMMA2-9B-INSTRUCT). We target medium-sized models in this work, which is of interest to post-training pipelines and end users with a modest amount of compute. Based on the CoT length, we choose the generation lengths (for zero-shot and RL) to be sufficiently large yet favor conciseness: 2000 tokens for LINEARDEPTH and PARTWHOLE, and 6000 tokens for KK.

### 4.1 ZERO-SHOT PERFORMANCE

Figure 2 shows the zero-shot performance of the models on the LINEARDEPTH, PARTWHOLE and KK datasets, Appendix Figure 4 shows the same for the response length and the fraction of completions that have the correct format. We observe that the defined difficulty measure is a good measure

of model performance: the accuracy generally declines linearly with increasing difficulty, possibly preceded by a sharp drop beyond the initial difficulties. Smaller models from the same model family generally perform worse than larger models. At a given model size, some model families are superior to others. The QWEN3 models perform the best across all datasets, G9B performs worse than L3B (except for KK where it performs slightly better). Comparing to the results reported in Xie et al. (2025) for QWEN2.5-7B-INSTRUCT-1M (zero-shot and finetuned), we obtain considerably better zero-shot performance with Q4B, despite using a smaller model. This may be explained by either it being a better model or the results being very sensitive to inference parameters such as temperature. The KK dataset is challenging for all models. Except for the QWEN3 models, all models have very low performance and zero performance for difficulty beyond 5. In the cases where the (average) response length does not increase linearly with difficulty, the model's accuracy drops significantly with difficulty, indicating that the model is not reasoning sufficiently. Generally, the model does not exceed the allocated response length. L1B and L3B have long reasoning traces for KK, but fail to produce correct results or follow the answer format. Looking at their reasoning traces, they engage in reasoning loops/overthinking (Sui et al., 2025) without real progress and repeat a short sequence of characters many times ($\sim 400$ times) before producing the EOS token. The drop in accuracy cannot be explained by the LLM failing to follow the answer format, as the fraction of completions following the correct format is relatively high, except for L1B, which can be attributed to the small model size resulting in fewer capabilities. According to the training reports (Yang et al., 2025; Touvron et al., 2023; Team et al., 2025a), the LLMs were trained on "synthetic" data distilled from the web and other LLMs, yet it is unclear whether they were trained on data generated by a synthetic data generator like in our setup. We see that all of the LLMs have some zero-shot capabilities, for low difficulties at least. The QWEN3 models were developed with reasoning capabilites in mind, and indeed perform the best.

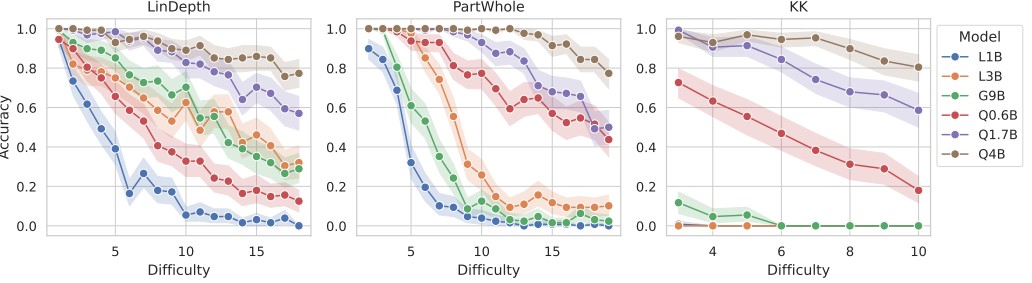

Figure 2: Zero-shot accuracy per dataset.

## 4.2 POST-TRAINING

We train on difficulties 1-5, 2-10 and 3-6 for LINEARDEPTH, PARTWHOLE and KK respectively. Section C.1 reports additional training details. We define the in-dist (in-distribution) accuracy as the average accuracy over the in-distribution difficulties, the ood (out-of-distribution) accuracy as the average accuracy over the out-of-distribution difficulties (calling the generalization to larger difficulties "out-of-distribution", as argued before). We define the generalization gap as the difference between the in-dist and ood accuracies (in-dist − ood). Figure 3 shows the ood accuracies at the final epoch for GRPO and SFT. Appendix B presents additional results for in-dist accuracies, the generalization gap, similar results for the response length and the fraction of responses with correct format at the final epoch, for both GRPO and SFT.

**Format-following:** First of all, we rule out format-following issues via Figures 6 and 10. For both in-dist and ood data, the fraction of responses with correct format is relatively high for all models and problem types. It is almost 100% for LINEARDEPTH and PARTWHOLE, and above 80% for KK. Note that format-following only affects RL training through a bad reward, but not SFT training.

**Impact of curriculum learning on RL (GRPO and PPO):** We generally see that standard random sampling (*Standard*) performs competitively for all models and problem types. There is no single curriculum that is always better. Moreover, the zero-shot performance is also relatively good for the larger QWEN3 models. For the LLAMA3.2 models, the *UpToDiff* and *SingleDiffDec* curricula perform worse than the other curricula, questioning the common use of this curriculum. The reward

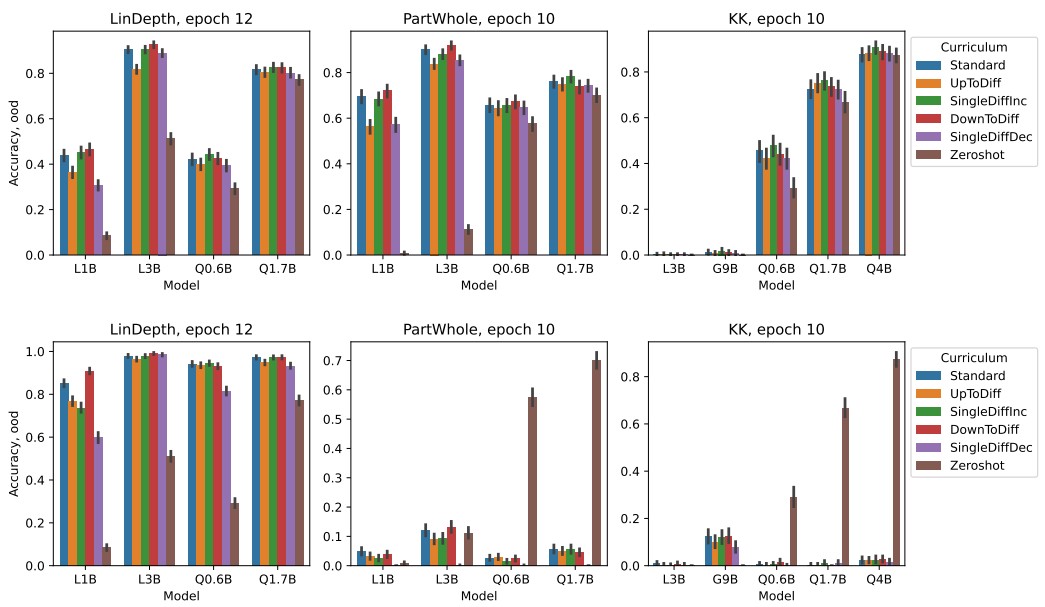

Figure 3: Out-of-distribution (ood) accuracies at the final epoch for GRPO (**top**) and SFT (**bottom**).

function provides an effective training signal as evidenced by the finetuned performance (except for L3B and G9B on KK), but the curriculum schedule does not increase the usefulness of the training signal, e.g., by first getting a training signal for easier tasks. The curriculum schedule also does not affect the response length distribution for GRPO, see Figure 7. In particular, this means that the model adapts its response lengths to the evaluation dataset rather than the last seen training dataset (which differs per curriculum schedule). Do the same results hold for PPO? PPO uses a critic model rather than just the rewards as the value function in GRPO. Appendix Figure 8 shows the accuracies at the final epoch for PPO for different curricula per difficulty, mirroring the GRPO conclusion that there is no clear advantage of any curriculum.

**Impact of curriculum learning on SFT:** For SFT, we similarly observe that the *SingleD-iffDec* curriculum performs the worst. For the other curricula, there is no significant difference, except for the very small L1B model. On the PARTWHOLE and KK datasets, the perfor-mance of the SFT-finetuned models breaks down, underperforming zero-shot significantly. Inspecting the reasoning traces for PARTWHOLE, we find that SFT-finetuned models lose their mathematical abilities, failing to correctly add a sequence of numbers as SFT tries to adapt all num-bers at once: `...there are 15 + 15 + 13 + 9 + 2 + 14 = 70 in total,` which sums up to 68 instead of 70. For KK, we find that SFT-finetuned models lose their logical abilities, reaching incorrect logical conclusions even for "easy" statements like `"Emma is a knight if and only if Emma is a knave,"` Owen claimed., incorrectly deducing that Owen is a knight (not lying). This does not happen as often for RL-finetuned models which explore various assumptions and backtrack. This happens right from the first epoch, see Figure 12. For KK, we cannot think of a straightforward educated remedy to make the CoT trace longer to approach the zero-shot response length distribution, since the CoT trace is already relatively verbose. Figure 9 reveals that the in-dist accuracy is non-zero whereas the ood accuracies are close to zero, pointing to memorization (Chu et al., 2025).

**RL vs SFT:** Summarizing, we generally see that RL performs better than SFT for in-dist and ood performance and the generalization gap. Whereas SFT-finetuned models lose their mathematical abilities, RL-finetuned models reason step-by-step, adding numbers one at a time. A reason could be that SFT imposes an unnatural reasoning style onto the model, so "the model memorizes, but does not understand". Interestingly, curriculum learning could alleviate this problem, but we found that it has no impact on performance. The SFT performance for PARTWHOLE could potentially be improved by adapting the CoT trace to perform the step-by-step addition rather than doing it in one step, as argued above. Could the difference between RL and SFT be explained by insufficient reasoning?

Figures 7 and 11 show the response length distributions for GRPO and SFT, respectively. We see that the distribution of response lengths is very similar between the RL-finetuned and zero-shot models (except for the small L1B), whereas SFT-finetuned models have significantly shorter response lengths. This is further supported by plotting the response length over time (Figures 12 and 13) and across curricula at the final epoch (Figure 14). We find that the CL schedule has minimal impact on response length, explaining why SFT performance is not impacted by the CL schedule. The response length drops significantly after the first epoch, but then stabilizes. For SFT, future work may use CoT traces that closely follow the model response length distribution and possibly decrease the CoT trace length over the course of training.

**Does "SFT memorize, RL generalize"?** We now discuss the paradigm "SFT memorizes, RL generalizes" suggested by Chu et al. (2025) in light of our results for the *Standard* curriculum (which we found to perform very similarly to other curricula). They show that SFT memorizes (with high in-dist accuracies, low ood accuracies), whereas RL generalizes (with low in-dist accuracies, high ood accuracies), and this effect becomes stronger for larger compute budgets (number of batches). More precisely, SFT performance is (significantly) lower/higher than RL performance on in-dist/ood data. While our experimental setup does not allow to measure the increase in this effect with respect to the training budget, we can assess whether the effect is present. We saw that SFT adopts the short response lengths from the CoT traces, but this does not mean it memorizes. Figures 5 and 9 show that the generalization gap is larger for SFT than for RL only on the PARTWHOLE dataset, whereas the opposite is true for LINEARDEPTH, and mixed for KK. Moreover, SFT underperforms RL on KK (see Figure 3) in terms of ood accuracy. We conclude that this paradigm depends a lot on the underlying dataset, the evaluation metric, and whether the finetuning method is successful or not. Our datasets test arithmetic and logical reasoning, with a precise definition of difficulty in terms of the number of reasoning steps, where we cannot confirm this paradigm.

## 5 CONCLUSION

Given that many of the real-world benchmarks like AMC, AIME are contaminated, we focused on synthetic reasoning tasks to assess whether post-training can improve reasoning capabilities of LLMs, and whether CL can boost performance. We found that CL consistently has no significant impact on accuracy and response lengths, across several datasets and model families. While small models are more impacted by the exact choice of curriculum, larger models are not affected as strongly. Moreover, SFT underperforms RL on some of the datasets and we identify response length as a potential reason. The CoT traces being significantly shorter than the natural response length distribution of the zero-shot model, SFT tries to impose its concise reasoning style onto the model, which results in poor performance. Interestingly, CL does not help in this regard, neither when starting with simple problems, paralleling the education curriculum, nor when starting with difficult problems, resembling the model's more verbose reasoning style. Our results should inform post-training practice that commonly starts with easy examples and adds more difficult examples gradually. For SFT, we postulate that CL may be more effective when the CoT traces initially reflect the model's reasoning style, in terms of length or other metrics. We plan to study this in future work by adapting the CoT trace to the model's reasoning style and implementing a curriculum, similarly to progressive finetuning (Liu et al., 2024). While results could improve by looking at pass@k accuracy, we judge it inadequate because it increases the burden on the end-user (having to inspect $k$ rollouts) and is more expensive. In fact, RL training can be seen as optimizing inference efficiency by internalizing pass@k. The model is optimized towards the the correct samples out of $k$ rollouts. SFT does not do this, but instead optimizes a single ground-truth answer, which may not be adapted to the model's reasoning style acquired during pre-training.

The conclusions are specific to our setting and may not hold for other settings such as different datasets with linguistic complexity, reward functions or finetuning strategies, larger or smaller models, larger dataset sizes or different curriculum schedules. Future work can consider more settings, or real-world datasets without data contamination.

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

CONTENTS

# A    DATASET EXAMPLES

| Problem Type | Difficulty | Problem | Reasoning Trace | Answer |
|---|---|---|---|---|
| LINEARDEPTH | 5 | Alexander has 20 liters of milk. Alexander has 16 liters of milk more than Matthew. Olivia has 19 liters of milk more than Matthew. Ava has 12 liters of milk fewer than Olivia. Ava then receives 10 more liters of milk from Abigail. Emma has 4 liters of milk more than Ava. How many liters of milk does Emma have? | Alexander has 20 liters of milk. Alexander has 16 liters of milk more than Matthew. So Matthew has 20 - 16 = 4 liters of milk. Olivia has 19 liters of milk more than Matthew. So Olivia has 4 + 19 = 23 liters of milk. Ava has 12 liters of milk fewer than Olivia. So Ava has 23 - 12 = 11 liters of milk. Ava then receives 10 more liters of milk from Abigail. So Ava has 11 + 10 = 21 liters of milk. Emma has 4 liters of milk more than Ava. So Emma has 21 + 4 = 25 liters of milk. | 25 |
| PARTWHOLE | 5 | Emma has 2 vases. Ava has 5 vases. Abigail possesses 7 vases. Olivia possesses 4 vases. Matthew has 6 vases. If everyone sums up the vases that they have, what is the number of vases that everybody has in total? | Emma has 2 vases. Ava has 5 vases. Abigail possesses 7 vases. Olivia possesses 4 vases. Matthew has 6 vases. If everyone sums up the vases that they have, there are 2 + 5 + 7 + 4 + 6 = 24 in total. | 24 |
| KK | 3 | A very special island is inhabited only by knights and knaves. Knights always tell the truth, and knaves always lie. You meet 3 inhabitants: Henry, Alexander, and Samuel. In Henry's words: "Samuel is not a knight". As Alexander put it, "Henry is a knave". Samuel told you that If Alexander is a knave then Henry is a knight. So who is a knight and who is a knave? | Let's think step by step, by considering whether each person is lying and if that leads to contradiction. Assume Henry is a knight. No contradiction is found in their claim that Samuel is not a knight. Samuel cannot be a knight, because this would contradict the claim of Henry that Samuel is not a knight. Samuel cannot be a knave, because this would contradict the false claim of their own that If Alexander is a knave then Henry is a knight. We have exhausted all possibilities for Samuel, so let us go back and reconsider Henry. Assume Henry is a knave. No contradiction is found in their false claim that Samuel is not a knight. Assume Samuel is a knight. No contradiction is found in their claim that If Alexander is a knave then Henry is a knight. Assume Alexander is a knight. No contradiction is found in their claim that Henry is a knave. This leads to a feasible solution. | Henry is a knave, Alexander is a knight, and Samuel is a knight. |

Table 2: Additional dataset examples.

# B    ADDITIONAL RESULTS

## B.1    ZERO-SHOT

Figure 4 shows the fraction of completions that have the correct format and response length, per dataset.

## B.2    GRPO

For GRPO, Figure 5 shows the in-dist accuracies and generalization gap at the final epoch, Figure 6 repeats the same for the fraction of responses with correct format and response length at the final epoch.

## B.3    PPO

For PPO, Figure 8 shows the accuracies at the final epoch for different curricula, per difficulty.

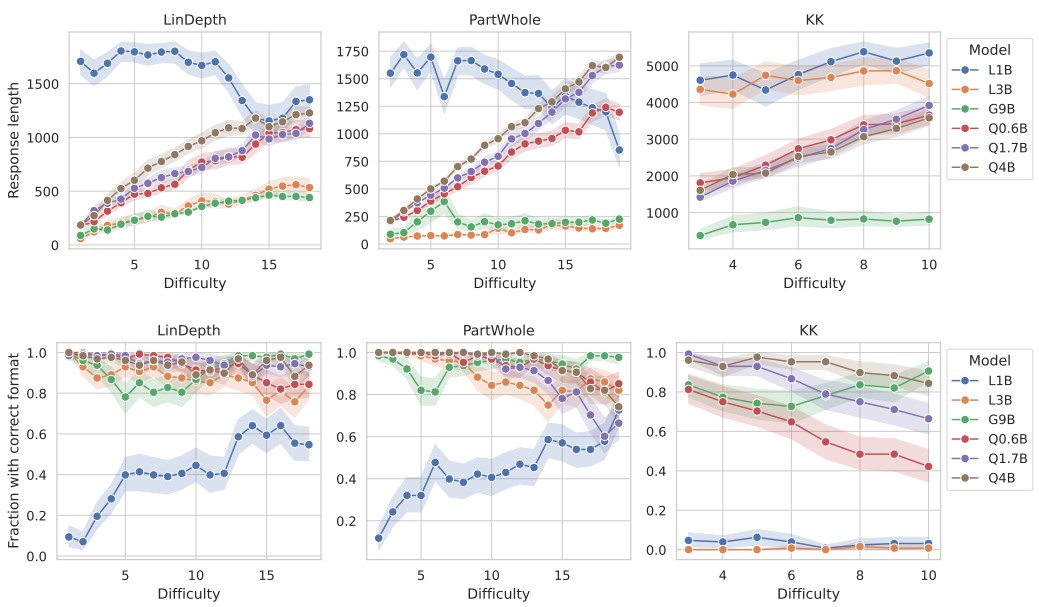

Figure 4: Zero-shot: Response lengths and fraction of completions that have the correct format, per dataset.

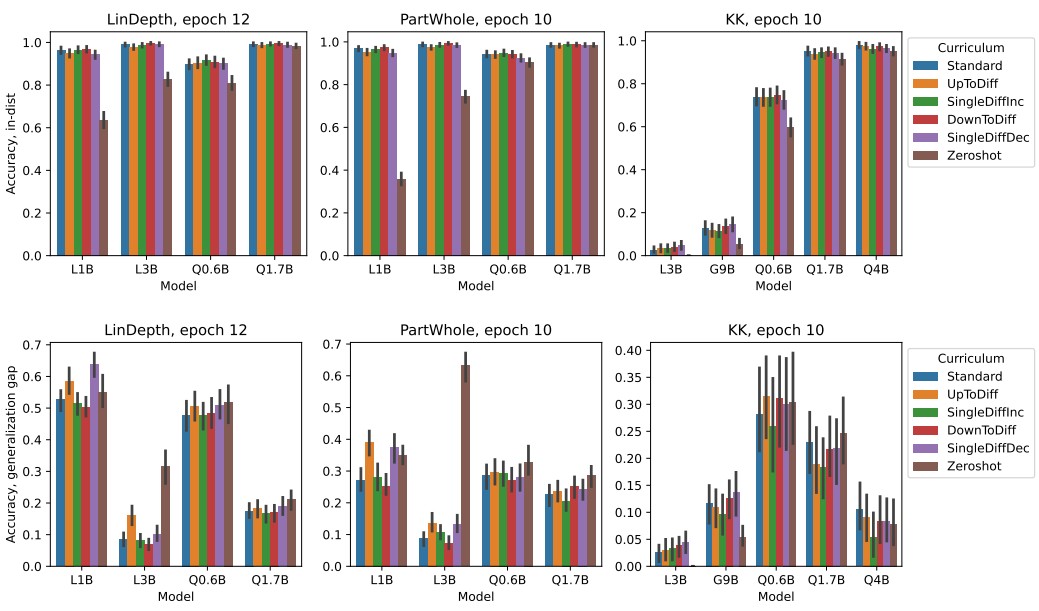

Figure 5: GRPO: in-dist and generalization gap of accuracies at the final epoch.

## B.4 SFT

For SFT, Figure 9 shows the in-dist accuracies and generalization gap at the final epoch, Figure 10 repeats the same for the fraction of responses with correct format and response length at the final epoch.

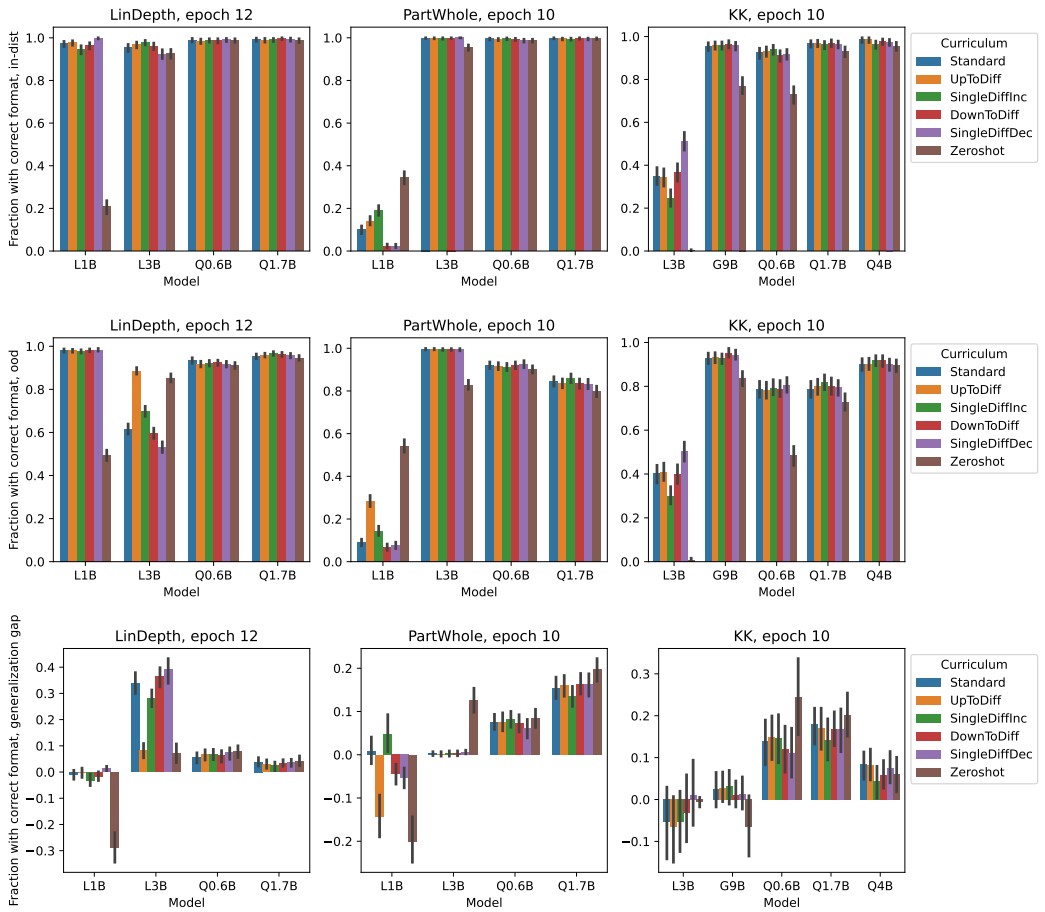

Figure 6: GRPO: Fraction of correct format for in-dist (in-dist), ood (ood) and generalization gap (in-dist − ood) at the final epoch.

### B.5 POST-TRAINING: GRPO VS SFT

Figures 12 and 13 compare GRPO against SFT over time in terms of accuracy and response length. Figure 14 compares GRPO against SFT across curricula at the final epoch.

## C IMPLEMENTATION DETAILS

We use the `VeRL` RL framework for all experiments and implement a curriculum sampler on top of it, adapting the code where necessary. We branched off commit `f0b4abaefc45573a591160896f8d544d8a34e45f` from `VeRL` (version `0.4.1.dev`). Due to active development of `VeRL`, we updated the SFT training code script to commit `3cc7695f4c70620ad871437037856f32182de096`. `VeRL` also supports zero-shot generation and SFT training, which we use to reduce differences in model performance due to different frameworks as much as possible. We extended the `VeRL` SFT training code to perform rollouts on the validation dataset during training. For performance reasons (as is the standard in `VeRL`), we filter out examples with more than 800 input tokens and 1200 input+CoT tokens (with respect to the QWEN3 tokenizer). This filters out a very small fraction of the examples. `VeRL` outputs the generated rollouts as json files, which we then parse to extract the reasoning traces and answers. All plots show the performance on the validation dataset, error bars are computed across bootstrap samples. We will make the training code available upon acceptance.

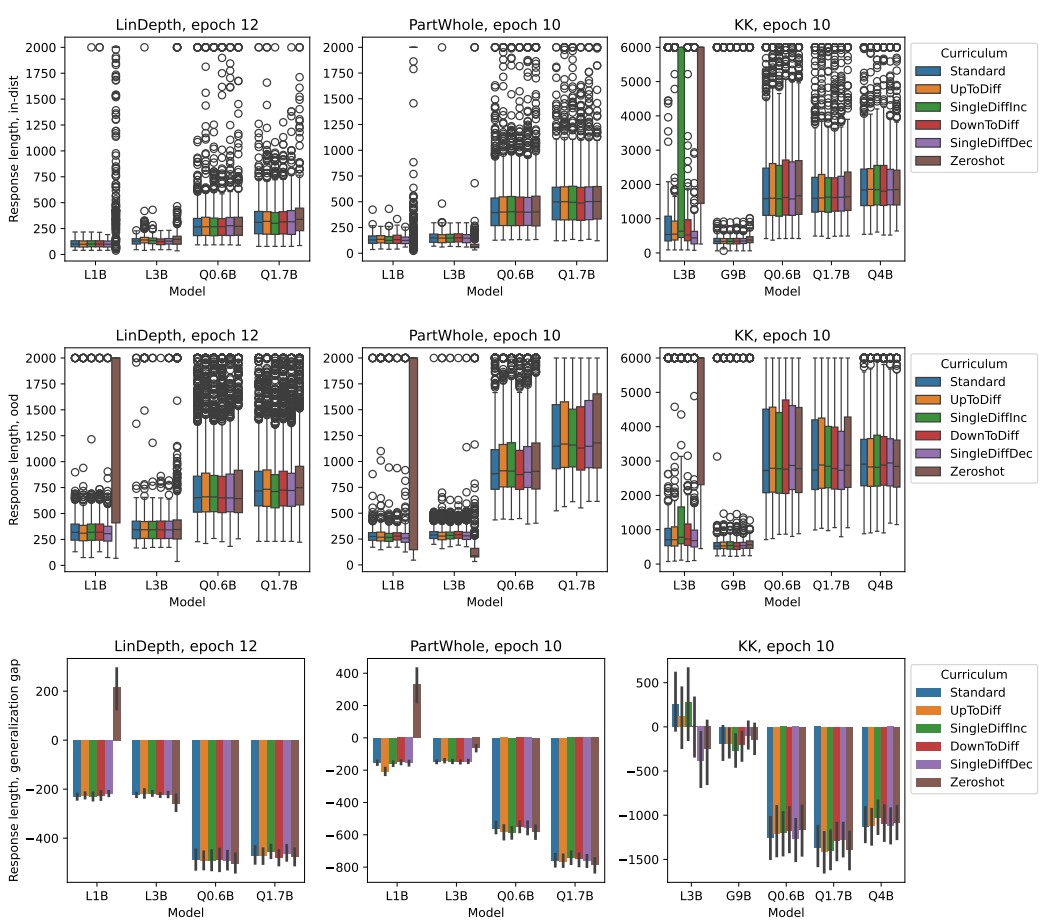

Figure 7: GRPO: Response length for in-dist (in-dist), ood (ood) and generalization gap (in-dist−ood) at the final epoch.

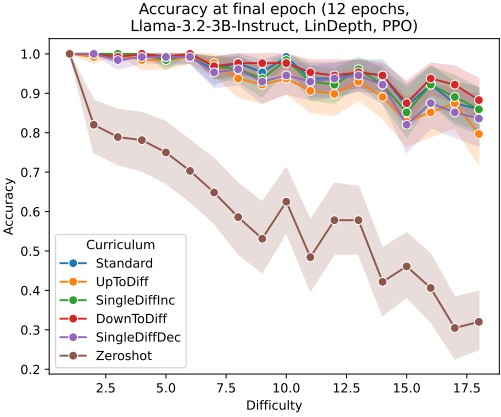

Figure 8: PPO: Accuracies at the final epoch for different curricula, per difficulty.

## C.1 HYPERPARAMETERS AND DATASET SETTINGS

We use the following hyperparameters for the different problem types:

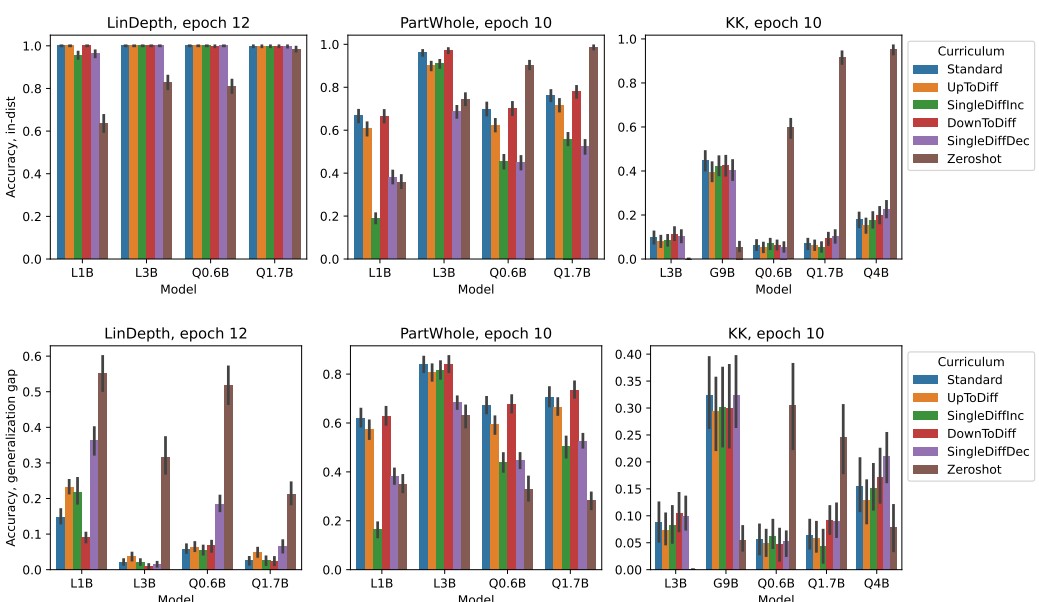

Figure 9: SFT: in-dist and generalization gap of accuracies at the final epoch.

- LINEARDEPTH (Opedal et al., 2025): train on difficulties $1 - 5$, evaluate on difficulties $1 - 18$, $m = 2$ epochs per curriculum phase, 4096 training samples per difficulty, global batch size 128 for SFT, global batch size 256 for RL, rollout length of 2000 tokens.

- PARTWHOLE (Opedal et al., 2025): train on difficulties $2 - 10$, evaluate on difficulties $2 - 19$ (difficulty 1 is excluded as it is a corner-case making the LLM think it is a trick question, especially during zero-shot), $m = 1$ epoch per curriculum phase, 4096 training samples per difficulty, global batch size 128 for SFT, global batch size 256 for RL, rollout length of 2000 tokens.

- KK (Xie et al., 2024): train on difficulties $3 - 6$, evaluate on difficulties $3 - 10$, $m = 2$ epochs per curriculum phase, 1024 training samples per difficulty, global batch size 32 for SFT, global batch size 256 for RL, rollout length of 6000 tokens.

We evaluate on 128 validation examples per difficulty. We adjust the micro batch size / token lengths per GPU per model to avoid OoM errors (which does not affect the result). All models are trained for one extra phase ($m$ extra epochs), repeating the last curriculum phase, i.e. $r = 1$.

Models are trained with `bfloat16` mixed precision. Following preliminary experiments, we use a learning rate of $1e - 4$ for SFT and $1e - 6$ for RL (except for RL on KK with learning rate $3e - 7$). Following common practice, we finetune SFT with LoRA (rank $r = 32$), whereas we do full finetuning for RL (which receives less gradient updates due to a larger batch size). RL uses temperature $T = 1$ during training rollouts. Following common practice, validation rollouts use temperature $T = 0$ resulting in deterministic rollouts. This ensures that the RL model rolls out the learned (greedy) policy rather than a stochastic variant. For comparability, we use the same temperature for SFT and zero-shot validation rollouts.

## C.2 OUTPUT PARSING AND REWARD FUNCTIONS

We detail the output parsing for the evaluation and the RL reward function. The model completions are parsed into *reasoning* and *answer* segments. Parsing first splits on a conclusion pattern that separates the reasoning from the answer, chosen among `</answer>`, `Final answer:`, or `</think>`, in this order. The thinking part is taken as all text before the first `</think>` (if present), while the answer part begins at the first `<answer>` tag (or defaults to the remainder if absent). The parsing fails if no conclusion pattern is detected, in which case the entire completion is treated as the answer.

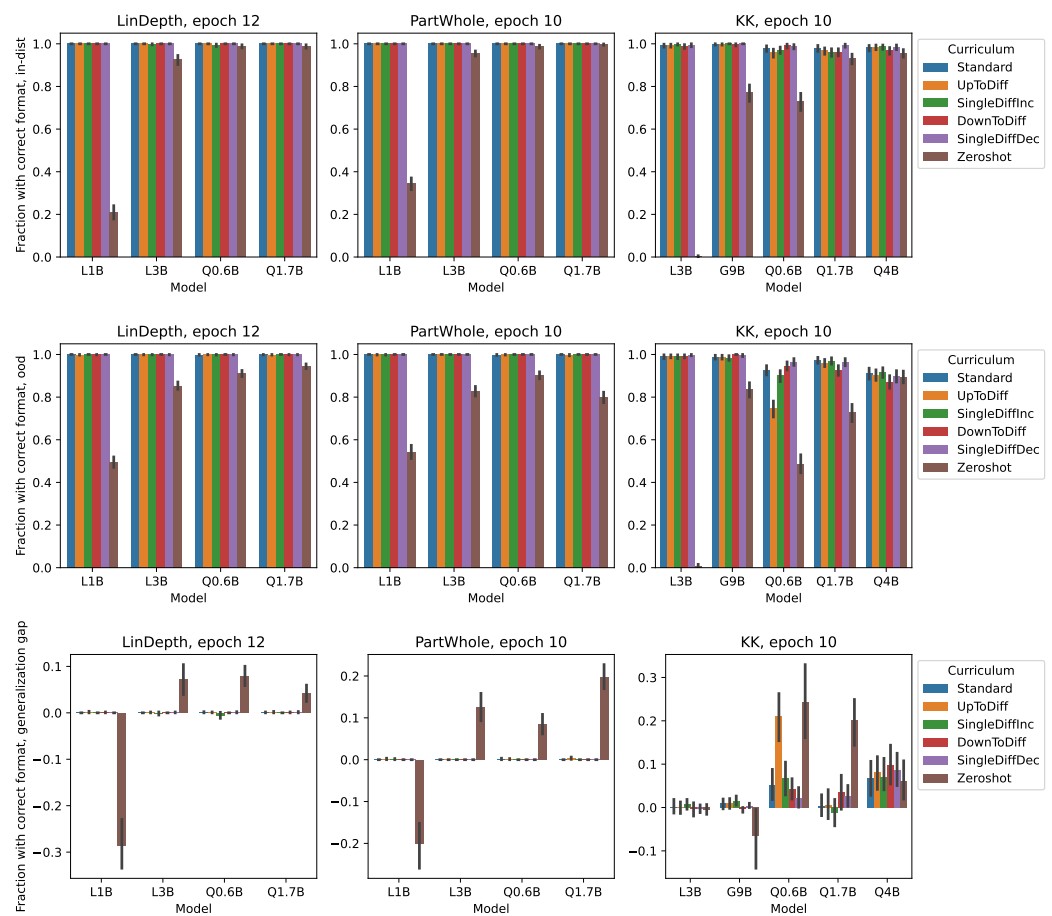

Figure 10: SFT: Fraction of correct format for in-distribution (in-dist), out-of-distribution (ood) and generalization gap (in-dist − ood) at the final epoch.

The thinking segment is ignored for evaluation, serving only to encourage reflection, and the answer segment is post-processed in a dataset-specific way, trying to match the expected answer format.

**Reward functions.** For MATHGAP, the reward is defined as:

$$R = \text{parsingSuccessful} + \text{isInt} + 4 \cdot \text{isCorrect},$$

We set $R = -2.0$ if the response hits the token limit.

For KK, we follow Xie et al. (2025) and define the reward as:

$$R = \text{formatScore} + \text{answerScore},$$

where formatScore is 1 if the completion has the correct format, else -1. The answer has the correct format if it contains the tags `</think>`, `<answer>`, and `</answer>` exactly once each, and in the correct order. The answer score is 2 if the answer is correct, else -1.5 if all characters appear in the answer, else -2. If the completion has the wrong format or the answer is empty, we set $R = -2.0$.

Although the exact weighting of components may influence performance, our preliminary experiments suggest that results are robust to moderate changes in the reward design.

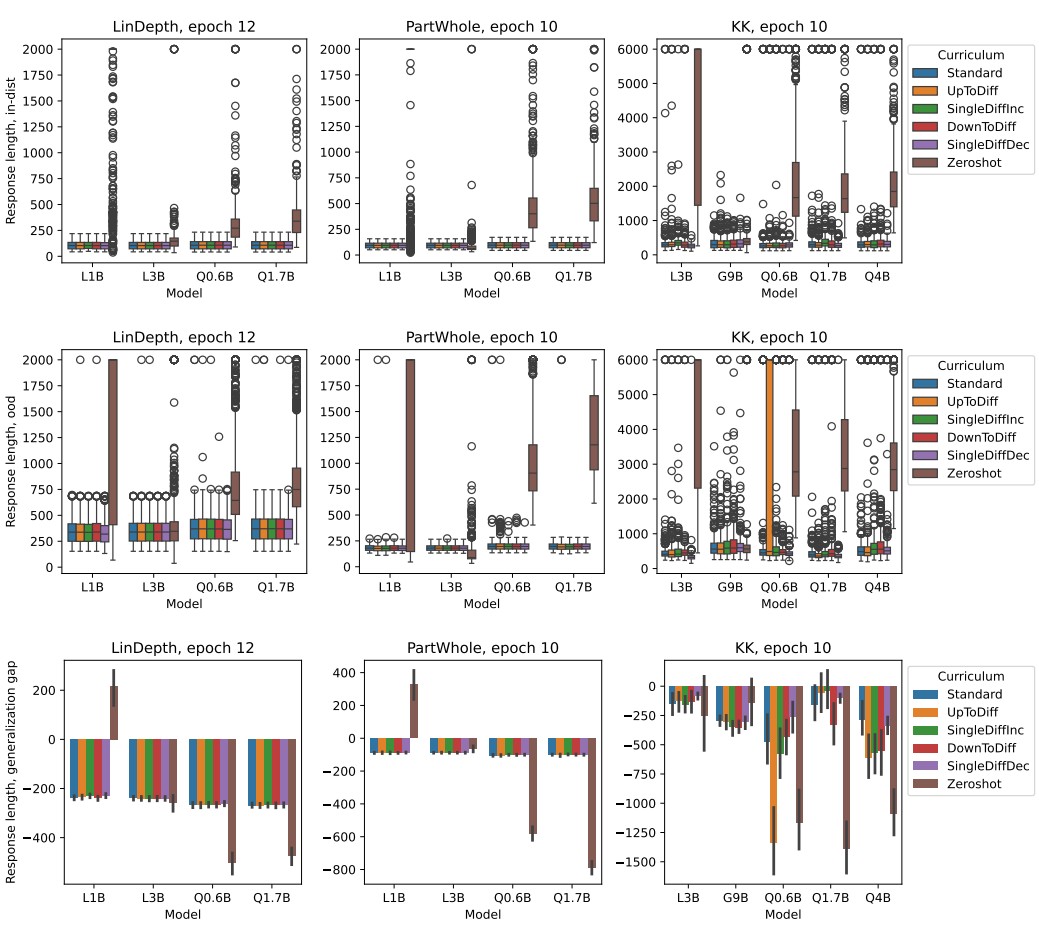

Figure 11: SFT: Response length for in-distribution (in-dist), out-of-distribution (ood) and generalization gap (in-dist − ood) at the final epoch.

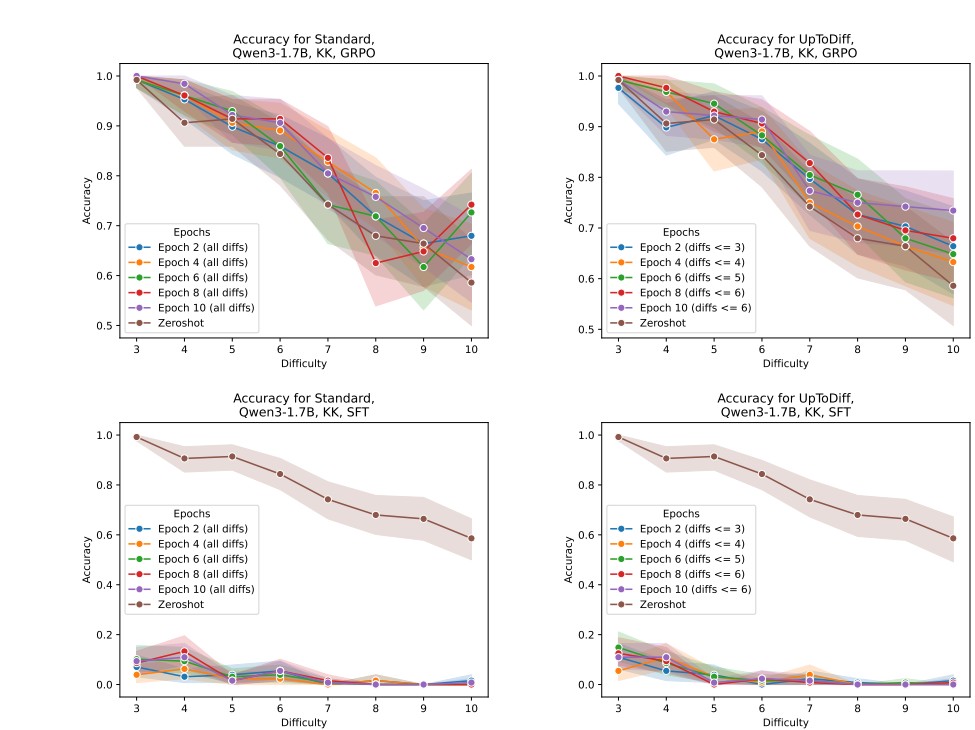

Figure 12: Accuracies over time across difficulty, GRPO (**top**) and SFT (**bottom**). We exemplarily plot the *Standard* (**left**) and *UpToDiff* (**right**) curricula for Q1.7B.

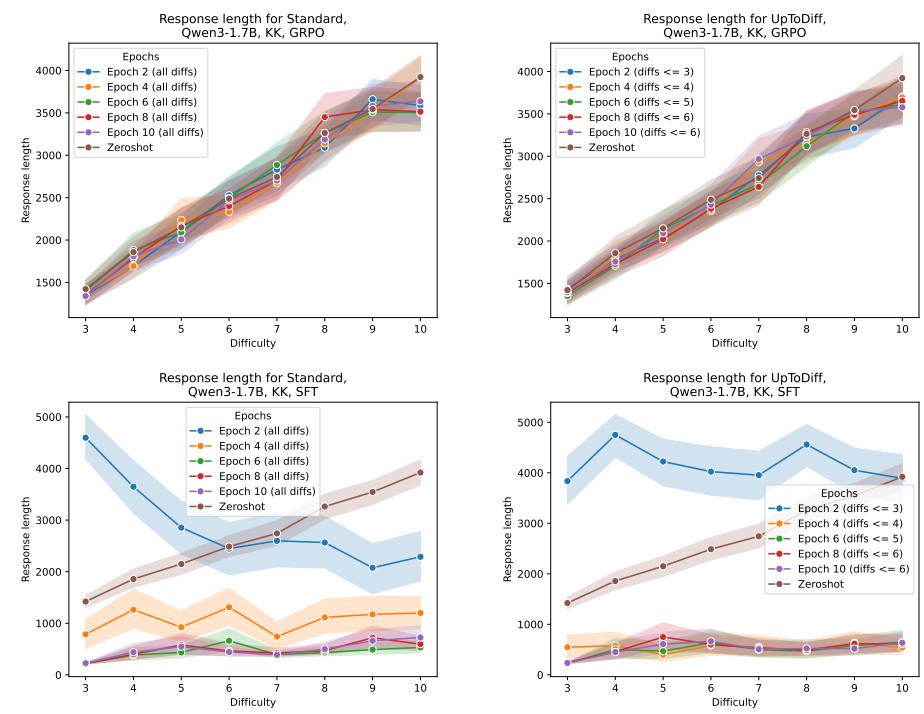

Figure 13: Response lengths over time across difficulty, GRPO (**top**) and SFT (**bottom**). We exemplarily plot the *Standard* (**left**) and *UpToDiff* (**right**) curricula for Q1.7B.

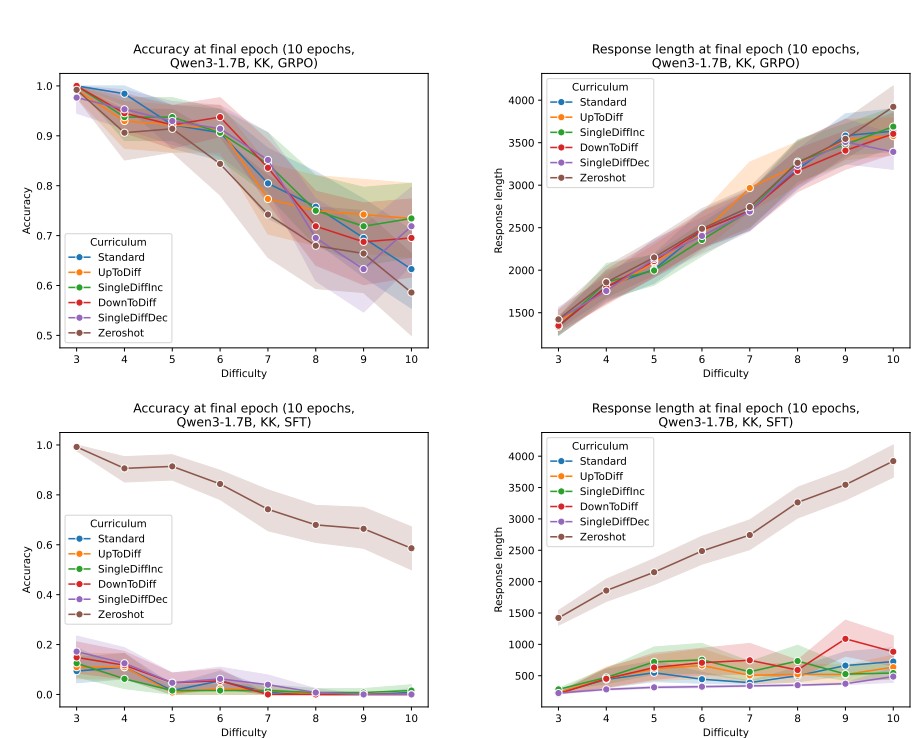

Figure 14: Accuracies and response lengths at the final epoch for GRPO (**top**) and SFT (**bottom**).

