# OpenReview forum: "On the Limits of Curriculum Learning for Post-Training Large Language Models"
_ICLR.cc/2026/Conference — Submitted to ICLR 2026_

### Official Review · Reviewer_71Fw · 2025-10-30

**Soundness:** 3
**Presentation:** 1
**Contribution:** 3
**Rating:** 6
**Confidence:** 2

**Summary:**

This paper investigates whether curriculum learning (CL) improves post-training performance of large language models on synthetic mathematical reasoning tasks. The authors systematically compare five curriculum schedules (including easy-to-hard, hard-to-easy, and random sampling) across multiple model families (LLaMA, Qwen, Gemma) and sizes (0.6B-9B parameters) using both supervised fine-tuning (SFT) and reinforcement learning (RL).

The main finding is that curriculum learning has no significant impact on performance—standard random sampling performs competitively with all curriculum schedules tested. The paper also observes that SFT can fail catastrophically on certain tasks (losing mathematical abilities), while RL maintains better performance, with response length identified as a key explanatory factor.

While the experimental design is sound and the research question is important, the paper suffers from significant writing quality issues that impede comprehension. Additionally, the use of synthetic datasets with minimal linguistic complexity and the limitation to medium-sized models raise concerns about the generalizability of the findings to real-world applications.

**Strengths:**

1. Important and Timely Research Question

The paper addresses a highly relevant question: whether curriculum learning, commonly used in LLM pre-training and post-training, actually improves performance on reasoning tasks. Investigating this widely-adopted technique with rigorous empirical validation is valuable, and reporting negative results (that curriculum learning may not help) is particularly important for guiding future research.

2. Sound Experimental Design

Despite the writing issues, the experimental design is well-conceived:

- Using synthetic datasets with controllable difficulty avoids data contamination issues that plague real-world benchmarks (AIME, GSM-8K, MATH)
- Systematic comparison of five curriculum schedules provides comprehensive coverage of common strategies
- Evaluation across multiple model families (LLaMA, Qwen, Gemma) and sizes (0.6B-9B) demonstrates generalizability
- Comparing both RL (GRPO, PPO) and SFT provides insights into how training paradigms interact with curriculum learning

3. Thorough Empirical Analysis

The paper provides comprehensive analysis including zero-shot baselines, in-distribution and out-of-distribution performance, response length and format-following behavior, and longitudinal tracking across training epochs. The identification of response length mismatch as a key factor in SFT failure is a concrete, actionable insight.

**Weaknesses:**

1. Writing Quality and Clarity Issues

Several sentences suffer from unclear or confusing constructions that hinder comprehension:

- Line 63: "performance typically degrades with reasoning length" is ambiguous. The preposition "with" lacks directionality—does performance degrade as reasoning length increases, or simply in the presence of reasoning length? This should be rewritten as "performance typically degrades as reasoning length increases."

- Lines 64-65: "Since CL acts on difficulty, there is reasonable expectation that it can improve performance on more difficult examples" presents an incomplete causal chain. Why does "acting on difficulty" necessarily lead to improved performance on harder examples? The mechanism linking CL's operation on difficulty to performance improvement needs explicit explanation.

- Lines 52-53: "none of the schedules we consider consistently improves performance" contains structural ambiguity. The placement of "consistently" between the relative clause "we consider" and the main verb "improves" creates confusion about what is being modified. Additionally, "consider" is inappropriate in an experimental context—"test," "evaluate," or "examine" would be more accurate. Suggested revision: "none of the curriculum schedules we tested consistently improves performance."

2. Incomplete Methodological Details

Section 3.3 (lines 298-307) provides only a high-level description of the RL reward function and training objective. For a paper centered on comparing training methods, the reward design is crucial and should be fully specified in the main text, not relegated to the appendix. Specifically:

- The exact reward function formulation with all components and their weights
- The loss function for both GRPO and PPO
- Justification for the chosen reward components and their relative importance

Without these details in the main text, it is difficult to assess whether differences between RL and SFT stem from the curriculum schedule or from the reward design itself.

3. Limited Experimental Scope with Overgeneralized Claims

The paper tests only medium-sized models (0.6B-9B parameters) but draws broad conclusions about curriculum learning in LLM post-training (e.g., Abstract: "CL has no significant impact"; Conclusion line 485: "CL consistently has no significant impact"). While the conclusion (lines 540-544) briefly acknowledges that findings are limited to the tested model sizes, this is a critical limitation that deserves more prominent treatment because:

- The paper does not discuss whether conclusions might differ for larger models (e.g., 14B, 32B) where curriculum effects might be more pronounced due to different capacity and training dynamics
- The strong claims in the abstract and introduction are not sufficiently qualified given this limitation
- For practitioners working with different model sizes, it is unclear how to interpret these results

Given that model scale is known to affect many aspects of LLM behavior, testing curriculum learning only at medium scales while making general claims about its effectiveness is a significant weakness. The abstract and introduction should be revised to explicitly state the scope, and the paper would benefit from at least preliminary experiments at one or two larger scales.

4. Representativeness of Synthetic Datasets

While using synthetic datasets to avoid contamination is laudable, the paper's emphasis on "minimal natural language complexity" (lines 58-59) raises serious concerns about generalizability. Although the conclusion (lines 540-544) acknowledges this limitation, we believe this is too important to dismiss with a brief mention in the conclusion:

- Real-world mathematical reasoning tasks involve substantial linguistic complexity (parsing word problems, resolving ambiguities, mapping natural language to formal operations) that may interact critically with curriculum learning effects
- The simplicity of the synthetic tasks might mask curriculum effects that would emerge in more naturalistic settings where models must learn both reasoning and language understanding simultaneously
- The paper does not provide sufficient discussion of why findings on minimal-language synthetic tasks should generalize to natural language reasoning, nor does it cite evidence that curriculum effects are invariant to linguistic complexity
- The design choice to minimize linguistic complexity, while methodologically convenient, may have inadvertently eliminated the very scenarios where curriculum learning provides benefits

The authors argue that synthetic tasks help "focus on reasoning ability" (line 59), but curriculum learning in real-world applications must address both reasoning and language complexity. At minimum, the paper should include experiments on at least one real-world dataset to validate that findings hold beyond the synthetic setting. Alternatively, a much more substantive theoretical or empirical discussion is needed to justify why the synthetic results should inform practice on natural language tasks.

5. Presentation Quality

Table 1 lacks a bottom horizontal rule (bottomrule in LaTeX booktabs), appearing unfinished and violating standard academic formatting conventions.

**I found the paper challenging to follow due to several instances of unclear and ambiguous wording and expressions. I strongly recommend that the authors carefully revise the manuscript to enhance clarity and readability.** Although I believe the paper's writing requires substantial revision, I find the experimental design to be generally sound. I am currently inclined to give a score of 6, but I remain somewhat uncertain and will adjust my rating based on other reviewers' feedback and the authors' responses during the discussion phase.

**Questions:**

see Weaknesses

---

### Official Review · Reviewer_Xhmm · 2025-10-31

**Soundness:** 2
**Presentation:** 1
**Contribution:** 2
**Rating:** 2
**Confidence:** 4

**Summary:**

The authors study the effects of curriculum learning (CL) applied to SFT and RLVR. The authors consider synthetic datasets which can be generated procedurally to avoid any data contamination. Each dataset has a notion of difficulty which is tied to the number of reasoning steps required to solve the logical or mathematical reasoning task. The authors finetune a wide range of SLMs including Llama3.2, Qwen3 and Gemma2 and show that no CL setups that they propose outperform random sampling.

**Strengths:**

* The use of synthetic datasets to control for dataset contamination.
* Thorough related works section.

**Weaknesses:**

* The presentation is rough and requires some development. The abstract reads like a summary of the experiments. The motivation for performing CL in the introduction is not clear. In terms of the writing all figure captions are not descriptive enough. The SFT and RLVR implementation details are not included in the main text e.g. did you use PEFT or full finetuning, what learning rate schedules did you use? Did you optimize hyperparameters for each dataset model pair?
* You make some claims which are not accompanied by experimental evidence or appropriate citations. For example:
    * Lines 31-32: “... and suggest that future work should explore alternative mechanisms for strengthening pure reasoning robustness in LLMs”. As far as I understand the robustness of the reasoning is not measured, but rather only whether the predicted answers match the ground truth answers in Figures 2 and 3?
    * Lines 38-40 “... consistent multi-step reasoning remains far from robust…”. I would argue that frontier LLMs are pretty good at multi-step reasoning. I can see from Figure 2 that only Qwen3-4b is robust to the problems under consideration, but to back up this claim you should also show the performance for frontier models e.g. GPT-4 or 5.
* The definition of difficulty is limited. The difficulty corresponds to the number of $d$ participants in the LinearDepth and PartWhole datasets. While for Knights and Knaves it corresponds to the number of characters introduced in the puzzle-like questions. There are other common definitions of difficulty—from the active learning literature for instance—which are not considered like using a high loss or uncertainty, or a low reward from an external reward model or using LLM-as-a-judge to score the difficulty of a problem.

**Questions:**

The main question I have is about the CL implementation. Every time you change the difficulty of the dataset you are introducing a distribution shift in the training process [1]. Did you have to make any particular changes to a regular training process to account for this? For instance, did you have a learning rate schedule per epoch since each epoch has data from a different difficulty [2]?

[1] Ash, Jordan T., and Ryan P. Adams. "On the difficulty of warm-starting neural network training." (2019).

[2] Lialin, Vladislav, et al. "Relora: High-rank training through low-rank updates." arXiv preprint arXiv:2307.05695 (2023).

---

### Official Review · Reviewer_DZss · 2025-11-01

**Soundness:** 2
**Presentation:** 2
**Contribution:** 2
**Rating:** 2
**Confidence:** 4

**Summary:**

This paper studies whether curriculum learning helps LLMs generalize in synthetic arithmetic and logical reasoning tasks, using carefully controlled difficulty levels. The answer is essentially "no": across SFT and RL or GRPO, curriculum schedules perform no better than standard random sampling. The authors identify response-length collapse as a key cause of SFT failure and note that RL preserves longer reasoning traces. Also notable is the effect of SFT on PartWhole and KK is significantly bad for the baseline models suggesting it is detrimental compared to zero shot reasoning.

**Strengths:**

1. Thorough literature review and scoping of the problem is appreciated.
2. Example dataset instances are also appreciated.
3. The authors study the performance of various small scale LLMs as a function of problem complexity and they find universal decline with problem complexity. What is this like for the SFT or curricula trained models?

**Weaknesses:**

1. Not enough novelty and depth of experimentation and insight. The work is a useful negative result, but the novelty is weak and the methodology is formulaic for this venue.
2. The experimental scope is also constrained: small model sizes and synthetic/toy settings that might not correspond to the real-world settings where curriculum learning might be more beneficial due to the complex nature of that setting.
3. The claim is framed as insightful, but most findings are rather unsurprising or task dependent: synthetic tasks with templated chain-of-thought offer limited room for nuanced curricula, and short SFT traces forcing unnatural brevity will obviously cripple reasoning and, at this point is quite clear from literature that longer CoTs help reasoning cf. deepseek R1 paper or from the theoretical viewpoint (takes tractable problem class from TC to NC cf. [this](https://arxiv.org/abs/2305.15408).

**Questions:**

see above

---

### Meta-Review · Area_Chair_tVyu · 2025-12-21

**Summary:**

The paper has four not positive reviews.

**Reviewer Concerns:**

Authors did not interact.

**Reviewer Scores:**

No possible change.

---

### Decision · Program_Chairs · 2026-01-26

Reject